# In Vivo Pollen Tube Growth and Evidence of Self-Pollination and Prefloral Anthesis in cv. Macabeo (*Vitis vinifera* L.)

**Francisco García-Breijo** [1] , **José Reig Armiñana** [2] , **Alfonso Garmendia** [3] , **Nuria Cebrián** [1], **Roberto Beltrán** [1] **and Hugo Merle** [1,*]

1   Departamento de Ecosistemas Agroforestales, Universitat Politècnica de València, 46022 Valencia, Spain; fjgarci@eaf.upv.es (F.G.-B.); nucego@hotmail.com (N.C.); robelmar@upvnet.upv.es (R.B.)
2   Instituto Cavanilles de Biodiversidad y Biología Evolutiva, Universidad de Valencia, 46010 Valencia, Spain; jose.reig@uv.es
3   Instituto Agroforestal Mediterráneo, Universitat Politècnica de València, 46022 Valencia, Spain; algarsal@upvnet.upv.es
*   Correspondence: humerfa@upvnet.upv.es; Tel.: +34-696658648

**Abstract:** Cultivar Macabeo is one of the most planted white grape varieties of northern Spain. A general agreement supports many *Vitis vinifera* cultivars possibly being self-fertile, although this seems to be a variety-dependent characteristic. No previous information about the mating system of cv. Macabeo was found. This study aimed to analyze its mating system and to compare the in vivo fertilization process with and without artificial cross-pollination. Two treatments were performed: emasculation and cross-pollination. The seed number was counted, and pollen tube growth was observed by microscopy. The results showed that cv. Macabeo is self-fertile and selfing probably occurs before the flower opens. Pollen was found over the stigma of flowers before capfall and ovule fertilization was observed even in emasculated flowers, which suggests that germination and pollen tube growth happened in a very early flower development stage. Cross-pollination increased the presence of the pollen tubes growing inside flowers but was not necessary for fruit set. Ovule fertilization was very fast as 24 h (h) were enough for pollen tubes to reach the end of stylar canals.

**Keywords:** self-pollination; cleistogamy; pollen tube; Macabeo; Viura

## 1. Introduction

Grapevine is one of the most important fruit crops with 7.15 million planted hectares and 79.1 million tons of world production in 2018 [1]. Macabeo, also known as Viura, is one of the most planted white grape varieties of northern Spain with nearly 32,000 ha [2]. It is also common in the Languedoc-Roussillon region in France [3] where it is called Macabeu.

Flower type, mating system and pollination mode have long since been discussed in *Vitis vinifera* L. [4–6]. A general agreement supports many *V. vinifera* cultivars possibly being self-fertile, but it seems a variety-dependent characteristic. For example, stamens in cv. Moscato Rosa are weak and curved, and artificial cross-pollination was necessary to obtain good yields [7]. Diminished fruit set and fewer seeds per berry appeared in the muscadine grape (*Vitis rotundifolia* Michx) with no effective cross-pollination [8]. For many cultivars, an increase in fertilization and, consequently, in fruit set has been reported when cross-pollination has been applied [9–11].

How and when pollen is transferred from anthers to stigmas has been discussed for *V. vinifera* cultivars. Some studies suggest that pollination takes place primarily by wind [12,13], while other

studies suggest that insects play an important role [5,14,15]. For several cultivars, however, direct pollen release before flower opening has been demonstrated [16]. The shedding of the calyptra is called capfall. Influence of temperature on flower development and capfall has also been studied [17,18].

True cleistogamy is defined as the formation of chasmogamous, open, cross-pollinated flowers and cleistogamous, obligate self-pollinated flowers on one individual, differing in floral morphology [19,20]. In *Vitis*, the term has been used *sensu lato* to refer to flowers that are pollinated and fertilized before flower opens, but without specific perianth, stamens and pistil modifications [5,8,21]. In this study, the terms "cleistogamous" and "cleistogamy" are used to refer to self-pollinated flowers prior to capfall, but not necessarily "true cleistogamy".

Several indications of selfing before capfall have been offered. Staudt (1999) showed that flowers of cv. Pinot Noir and cv. Muller-Thurgau had pollen on stigmas before capfall and pollen tube growth had already started, which suggests that both varieties were at least partially cleistogamous. In the flowers of cv. Barbera, viable pollen was also present on stigmas before capfall [22]. Heazlewood and Wilson (2004) confirmed the presence of pollen on the stigma of cv. Pinot Noir before capfall. By studying that pollen, these authors did not find any pollen tube growth before capfall. They concluded, conversely to Staudt, that fertilization should occur after the shedding of the calyptra. Cleistogamy does not appear in all cultivars, since mechanisms may be present inside the cap to prevent it [16].

Very little information about the mating system of most cultivars is available [16,21,23]. Although in vitro pollen quality and germination have been studied in quite a few varieties [24–26], the in vivo fertilization process has been scarcely studied [21,23]. To our knowledge, there is no information about the mating system of cv. Macabeo or the time needed for the pollen tube, from pollen germination on the stigma, to enter into the ovule to fertilize the egg cell. Therefore, this study aimed to analyze the mating system of cv. Macabeo and to compare the in vivo fertilization process with and without artificial cross-pollination.

## 2. Materials and Methods

### 2.1. Experimental Site

Research was carried out in June 2019 flowering season on Macabeo (Viura) in a commercial vineyard in Casas Ibañez (Albacete, Spain, 39.331037 N, −1.478663 W) (Figure S1a). The general site climate is hot-summer Mediterranean (Csa in Köppen-Geiger classification), with a long-term average annual rainfall of 417 mm and an average annual air temperature of 13.8 °C (w.s. 39.272222 N, −1.460278 W 773 masl). The plot's soil type is calcareous, with gravel and red clays.

### 2.2. Treatments

For the experiments, six grape bunches were randomly selected. The entire experiment was repeated in two healthy vines. Both vines were treated separately with just a few days difference. Of the six bunches per vine, three were used to assess the seed number by allowing berries to develop until maturity, and three were used for the flower sampling for microscopy. Two treatments were run on the 12 selected bunches: emasculation and cross-pollination. Both treatments were performed in each grape-bunch, but in a different subcluster. Two subclusters were selected in each bunch with around 20 young flowers per cluster and were then randomly assigned to treatment. Therefore, around 240 flowers were used for each treatment. The flowering stage of the selected flowers was around 24 h (h) before natural capfall. In the treated subclusters, the flowers that were too young or already without calyptra were removed immediately before treatment (Figure S1b,c).

For both treatments, first the calyptra was gently removed. Then, for the emasculation treatment, anthers were cut, and flower clusters were bagged in semi-permeable nylon bags until fruit set or flower sampling (Figure S1). The semi-permeable nylon bags prevented pollen from arriving by wind or pollinators. The emasculated flower clusters were bagged immediately after treatment and were never opened before evaluations (Figure S1d). For the cross-pollination treatment, after calyptra



removal, the flower clusters were bagged, and 24 h later, flowers were pollinated with pollen from another vine. For this purpose, the bag was removed, and flowers were brushed gently against those flowers collected at anthesis from a different vine. After pollination, flower clusters were rebagged until fruit set or flower sampling.

## 2.3. Seed Number

In September 2019, all the grapes from both treatments were collected and cut to count the number of seeds. One hundred and one and 87 berries were evaluated for cross-pollination and emasculation, respectively.

## 2.4. Flower Sampling for Pollination Experiments and Floral Structure Analysis

Six flowers per grape cluster and treatment were collected 48 h after calyptra removal. For cross-pollination experiment flowers were fixed 24 h after pollination, for emasculation treatment 48 h after anthers removal. All flowers were sampled early in the morning and immediately fixed in Formalin-Aceto-Alcohol (FAA) solution (10 mL formaldehyde 37%, 50 mL ethyl alcohol 95%, 5 mL glacial acetic acid; 35 mL of water) or Karnovsky [27]. Vials with fixed material were transported to the laboratory in an insulated container, and subsequently stored at 4 °C in the refrigerator until analyzed.

For autofluorescence-stained sections, flowers were fixed in FAA solution and were sectioned longitudinally on ca. 20 μm with a freezing microtome (Leica CM 1325), sections were stained with aniline blue fluorochrome (0.1% in PBS buffer) for 15 min (min), subsequently washed with distilled water and mounted in a microscope slide for observation under an epifluorescence microscope.

For semi-thin sectioning, flowers were fixed in 2% Karnovsky fixative for 2 h at 4 °C before they were washed 3 times with 0.01 M PBS, pH 7.4, for 15 min each [27]. Samples were dehydrated at room temperature in graded series of ethanol, starting at 50% and increasing to 70%, 95% and 100% for no less than 20–30 min at each step, embedded in Spurr's resin according to the manufacturer's instructions. Flowers were sectioned on 1–2 μm using a diamond knife (DIATOME Histo 45) and an ultramicrotome (Ultratome Nova LKB Bromma). Sections were stained with 0.1% toluidine blue.

Sections were examined with light microscopy (LM) and epifluorescence microscopy (EFM) to analyze stigma, style, ovary, ovules, pollen grains on stigma. All the LM and EFM observations were made by an Olympus Provis AX 70 fluorescence microscope equipped with an Infinity 2–3 C Lumenera® digital camera and analyzed with the "Infinity Analyze" software, v.6.4.1. For fluorescence microscopy, an Olympus U-ULS 100 HG epifluorescence system with a U-MWBV cube (excitation filter 400–440 nm, dichroic mirror 455 nm, barrier filter 475 nm) was used. For flower morphology observation, a Leica M165 stereomicroscope with a high-resolution IC80HD digital image capture system controlled by the LAS program was used.

## 2.5. Statistical Analysis

Statistical analyses and figures were made using R language [28] with RStudio [29]. In addition, agricolae [30] and ggplot2 [31] packages were used for post-hoc analyses and graphs, respectively. The average, standard error, skew, and kurtosis were assessed per treatment. For the means comparisons, the ANOVA test did not meet the normality of residuals requirement and, therefore, the non-parametric Kruskal–Wallis test was used. Normality of residuals was tested using Shapiro–Wilk test. The effect of experiment repetition was checked by the non-parametric Kruskal–Wallis test, within the treatments between vines.

## 3. Results

### 3.1. Analysis of the Mean Seed Number per Grape and Fruit Set

A nonsignificant difference in the mean seed number per grape was found between treatments (Figure 1; Table 1). The emasculation treatment produced slightly more seeds (1.425 + −0.078) than the

cross-pollination treatment (1.386 + −0.072), but this difference was not significant (Table 1). The effect of experiment repetition on the seed number was checked and no significant differences were observed (Kruskal–Wallis *p* value = 0.082).

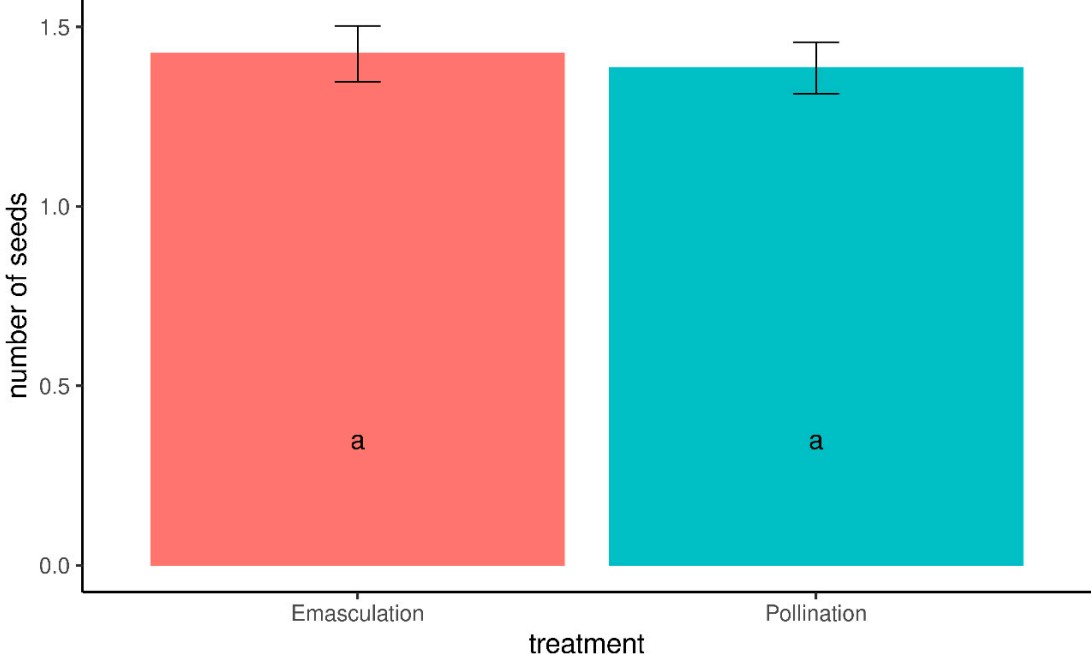

**Figure 1.** Barplot for the effect of treatment on seeds per grape. The columns with the same letter do not significantly differ from one another at $p \leq 0.05$, Df = 187, KW *p* value = 0.81.

**Table 1.** Mean seed per grape according to treatment.

| Treatment | N | Mean | HSD | KW | SE | Skew | Kurtosis |
|---|---|---|---|---|---|---|---|
| Emasculation | 87 | 1.43 | a | a | 0.08 | 0.45 | −0.03 |
| Pollination | 101 | 1.39 | a | a | 0.07 | 0.25 | −0.09 |

N, number of assessed berries; HSD, honestly significant difference; KW, Kruskal–Wallis test for the effect of groups on the mean seed number, *p*-value = 0.811 (Df = 187). The treatments with the same letter do not significantly differ from one another at $p \leq 0.05$; SE, standard error; Skew, skewness.

Both treatments behaved mainly the same when analyzing the distribution of seeds per grape. The number of grapes with 0, 1, 2 and 3 seeds were similar for both treatments (Figure 2). That is, the emasculation treatment had no effect on either the total seed number or seed distribution per grape.

Fruit set was not specifically examined in this experiment. Therefore, no exact initial number of treated flowers in each cluster was recorded. Nevertheless, each treatment was applied in six similarly sized flower clusters with around 20 flowers per cluster. No significant difference in the mean number of grapes obtained per cluster was found between treatments (Figure 3; Kruskal–Wallis *p* value = 0.571). This result suggests that emasculation had no negative effect on fruit set compared to cross-pollination. Although the data indicated no effect of treatment on fruit set, the experimental design was not intended for this purpose. Therefore, the result should be cautiously taken.

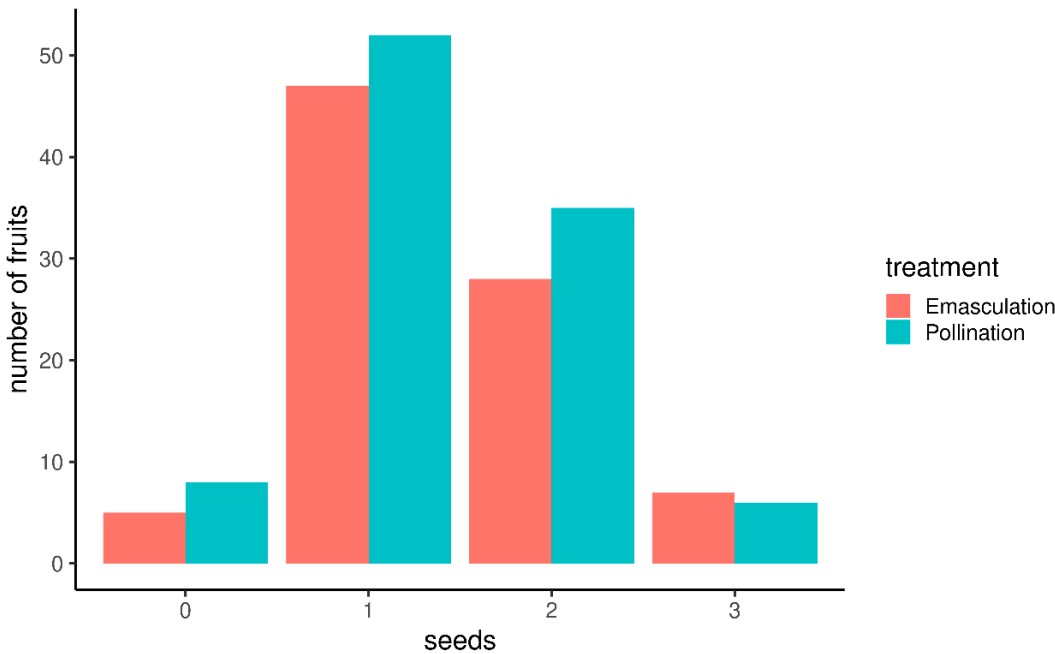

**Figure 2.** Barplot for seeds according to 0, 1, 2 and 3 seeds per grape per treatment. Dotted lines represent the mean value.

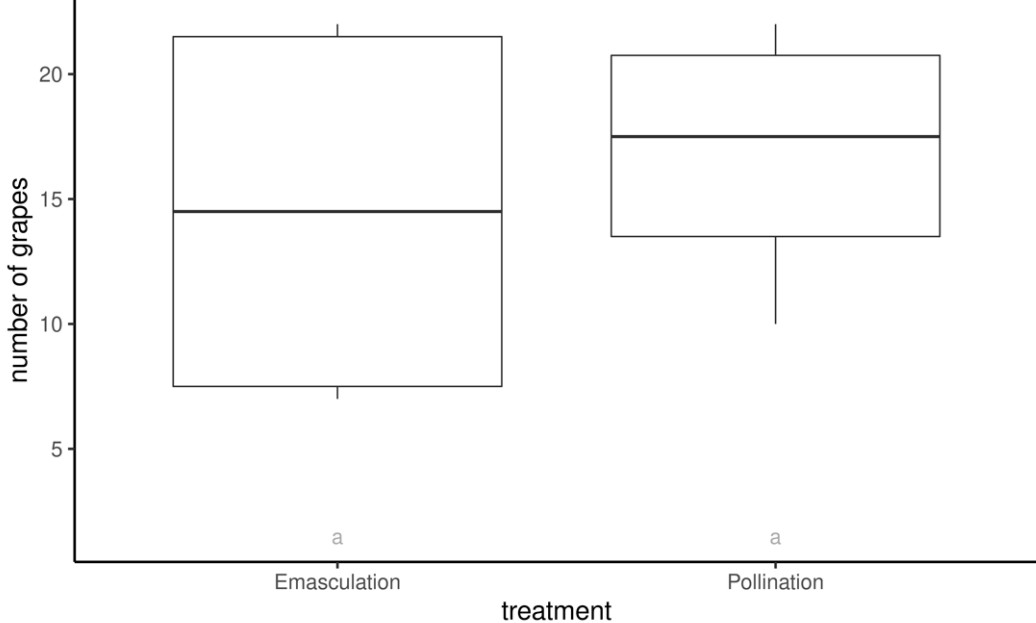

**Figure 3.** Boxplot for the effect of treatments on number of grapes. Boxes show the 25th and 75th percentiles. The lines in the boxes denote median values. The columns with the same letter do not significantly differ from one another at $p \leq 0.05$, Df = 11, KW $p$-value = 0.5711073.

### 3.2. Macabeo Flower Structure

All the observed Macabeo flowers were hermaphrodite with five sepals, five petals, five stamens and one pistil (Figure S2). Female flowers were absent.

The Macabeo stigma was wet and papillate (Figure 4a,b and Figure S3a). The densely arranged papillae were long, multicellular, uniseriate and uniform in diameter (Figure S3b). Papillae were covered with a cuticle lining and a film formed of secretory material, as revealed by toluidine blue staining (Figure S3b). The style comprised an epidermis layer, a parenchymatic tissue and a central

transmitting tissue (Figure 4a–c). The epidermis was composed of a single layer of cells rich in tannins covered by a thick cuticle (Figure 4c). Adjacent to the epidermis were 12–18 layers of vacuolated cortical cells (stylar cells). In these cells, the structures containing raphides were present (Figure S3a,c). Raphides were more numerous near the stigma, but also appeared on ovary walls (Figure 4a).

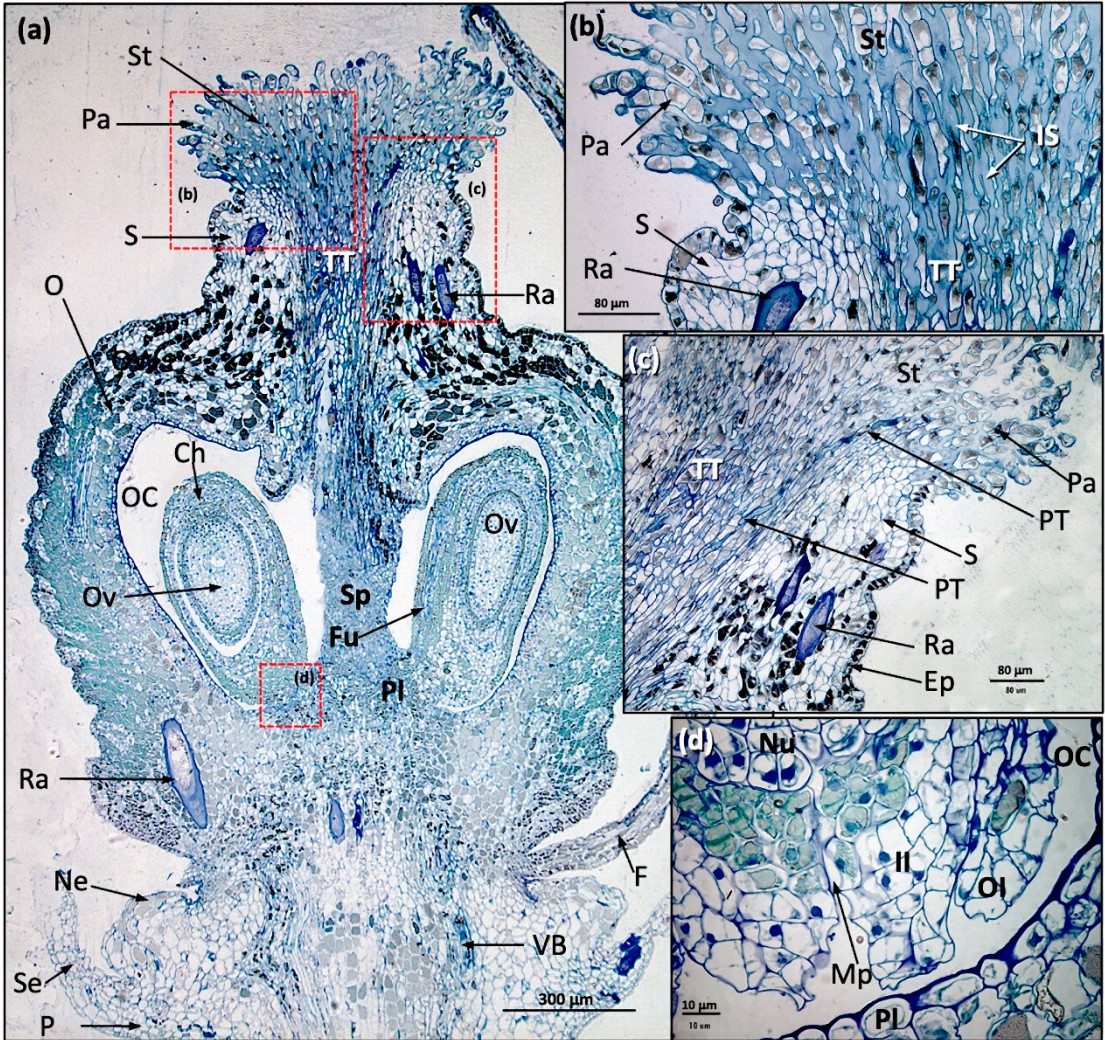

**Figure 4.** Longitudinal toluidine blue-stained semi-thin section (1–2 µm) of a Macabeo flower. (**a**) Flower structure, the details (red squares marking) are magnified: (**b**) papillae on stigma and transmission tissue. (**c**) Pollen tube growing through the transmission tissue. (**d**) Ovule micropyle zone. Abbreviations. Ch, chalaza; Ep, epidermis; F, filament; Fu, funicule; II, inner integument; IS, intercellular space; Mp, micropyle; Ne, nectary; Nu, nucellus; O, ovary; OC, ovary cavity; OI, outer integument; Ov, ovule; P, peduncle; Pa, papillae; Pl, placenta; PT, pollen tube; Ra, raphide; S, style; Se, sepal; Sp, septum; St, stigma; TT, transmission tissue; VB, vascular bundle.

Vascular bundles were present in the parenchymatic tissue, but only in the proximity to the ovary. The cylindrical transmitting tissue was located in the center of the style and was continuous with papillae (Figure 4a,b). In the mature gynoecium, the transmission tissue cells were arranged loosely and were surrounded by large intercellular spaces filled with secretion products (Figure 4a–c). These spaces progressively increased toward the ovary, where the transmission tissue cells seemed to form an area filled with secretion (Figure 4b).

### 3.3. In Vivo Pollen Tube Growth and Ovule Fertilization

At least nine flowers per treatment were studied by EFM. The emasculated flowers presented pollen on the stigmatic surface, but not large quantities (Figure 5a,b). These flowers were emasculated before capfall and then bagged. Thus, the most likely origin of pollen would be from its own anthers, which would have opened before bloom. In some emasculated flowers, pollen tubes were observed along stylar canals and entered the micropyle to cause ovule fertilization (Figure 5c,d and Figure 6). These flowers were fixed with FAA only 48 h after emasculation, by which time some ovules were already fertilized.

When cross-pollination was carried out, many pollen grains on the stigmatic surface and profuse pollen tube development in the stigma and style tissues were observed (Figure 7a,b). In addition, the mass growth of pollen tubes along the transmission tissue (TT) was observed (Figure 7a,c). In some flowers, the mass of pollen tubes reached the base of stylar canal (Figure 7a), while in other flowers the pathway of pollen tubes was shorter, finishing in the middle of the stylar canal (Figure 7c). In some pollinated flowers, pollen tubes entered through the micropyle into the ovule, releasing sperm cells to fertilize an egg cell and the central cell of female gametophyte (=embryo sac) (Figure 8). These flowers were fixed only 24 h after pollination, but the tubes in the micropyle could correspond to previous pollen grain germinations from their own anthers, as in the emasculated treatment. Profuse tube development on the stigma and style was only observed for the cross-pollination flowers, but not for the emasculated flowers. These observations point to a very fast fertilization process as the mass of tubes reached the vicinity of the ovule in only 24 h, and to a fertilization improvement due to a significant increase in the number of pollen tubes growing inside the flower.

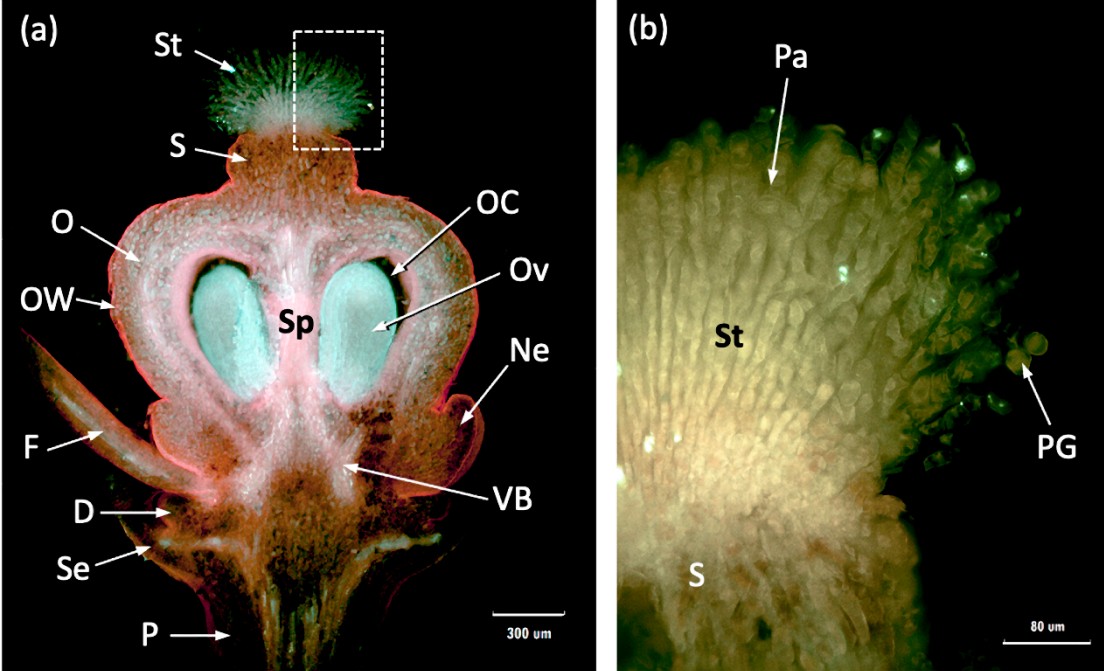

**Figure 5.** *Cont.*

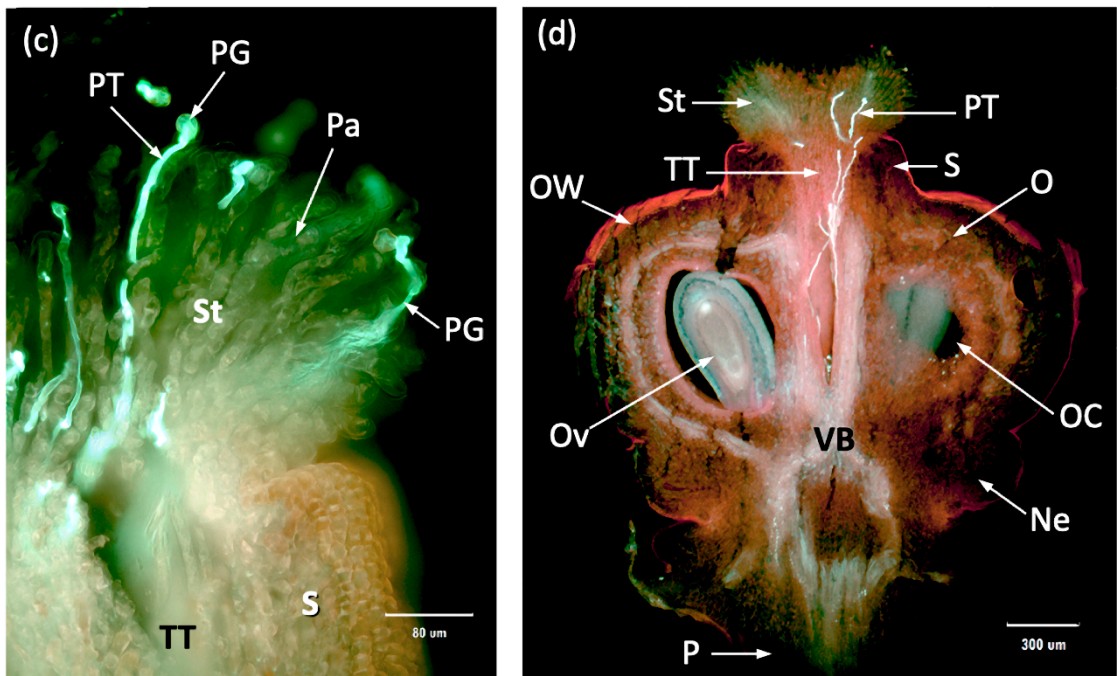

**Figure 5.** Longitudinal blue violet autofluorescence-stained freeze section (20 μm) of an emasculated flower. (**a**) Flower structure, visible pistil with stigma and style, ovary with ovules. (**b**) Papillous stigma, visible ungerminated pollen grains. (**c**) Germinating pollen grains, pollen tubes on stigma and in style. (**d**) Pollen tubes reach the stylar canal's base. Abbreviations. D, discus; F, filament; Ne, nectary; O, ovary; OC, ovary cavity; Ov, ovule; OW, ovary wall; P, peduncle; Pa, papillae; PG, pollen grain; PT, pollen tube; S, style; Se, sepal; Sp, septum; St, stigma; TT, transmission tissue; VB, vascular bundle.

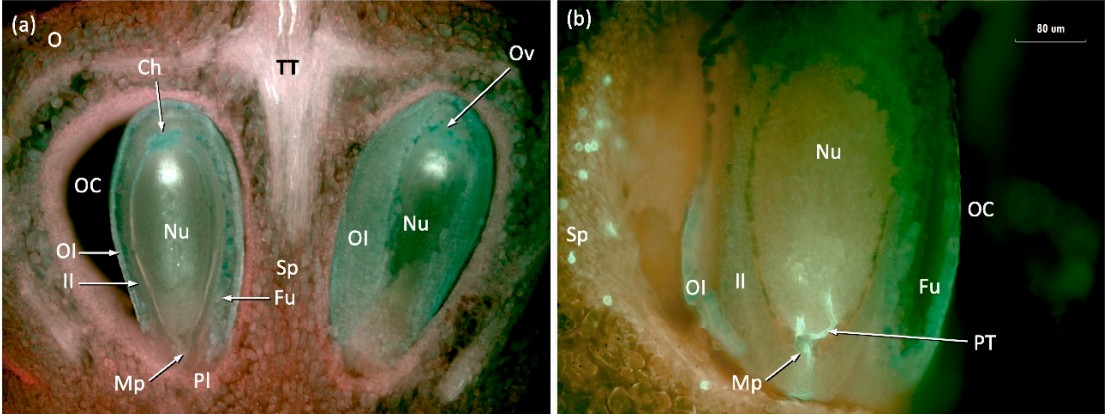

**Figure 6.** Longitudinal blue violet autofluorescence-stained freeze section (20 μm) of an emasculated flower. (**a**) Two mature ovules inside the ovary. (**b**) Pollen tube penetrating through the micropyle into the embryo sac. Abbreviations. Ch, chalaza; Fu, funicule; II, inner Integument; Mp, Micropyle; Nu, nucellus; O, ovary; OC, ovary cavity; OI, outer integument; Ov, ovule; Pl, placenta; PT, pollen tube; Sp, septum; VB, vascular bundle.

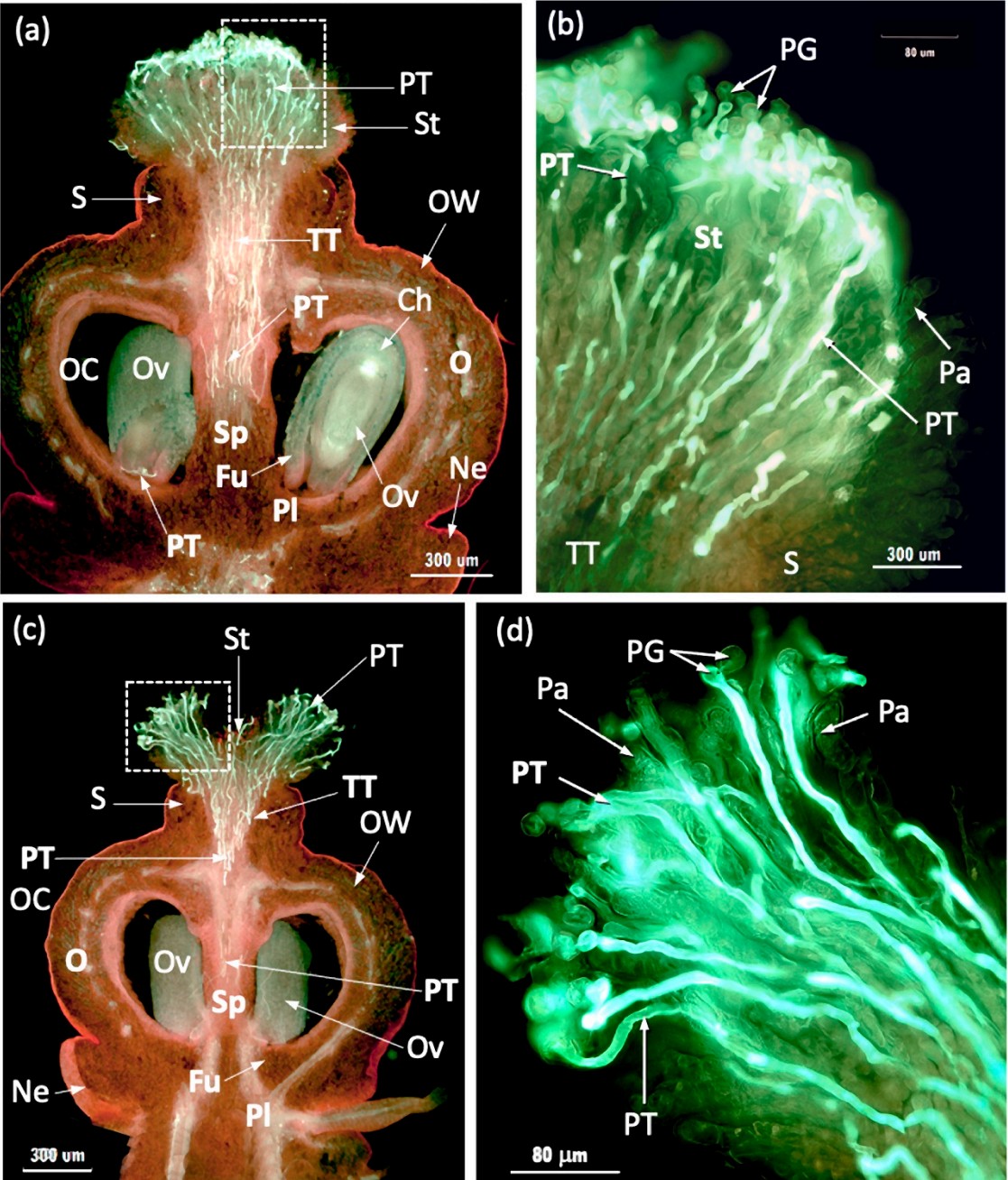

**Figure 7.** Longitudinal blue violet autofluorescence-stained freeze section (20 μm) of a cross-pollinated flower. (**a**) Pistil with visible abundant germinating pollen grains (magnified stigma in (**b**)), pollen tubes penetrating transmission tissue of the style and inside the ovule. (**c**) Pistil with visible abundant germinating pollen grains (magnified stigma in (**d**)), most pollen tubes reaching half of the stylar canal. Abbreviations. Ch, chalaza; Fu, funiculus; Ne, nectary; O, ovary; OC, ovary cavity; Ov, ovule; Pa, papillae; PG, pollen grain; Pl, placenta; PT, pollen tube; S, style; Sp, septum; St, stigma; TT, transmission tissue; VB, vascular bundle.

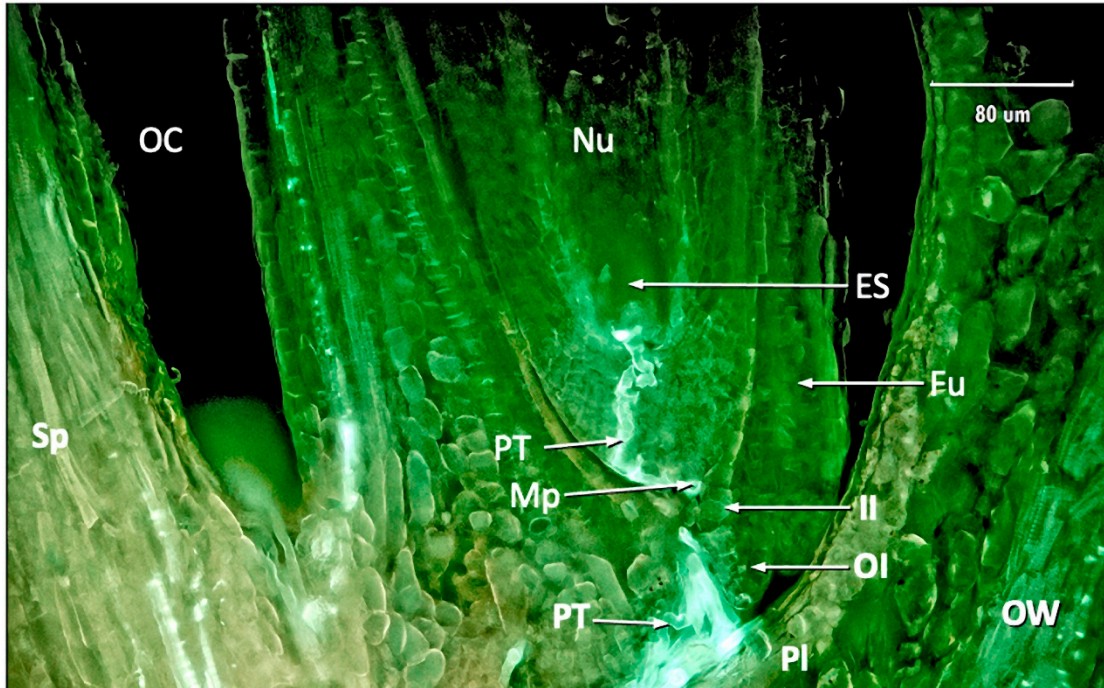

**Figure 8.** Longitudinal blue violet autofluorescence-stained freeze section (20 μm) of a cross-pollinated flower. Detail of the fertilization process of a mature ovule. The entrance of a pollen tube in the embryo sac through the micropyle. Abbreviations. ES, embryo sac; Fu, funiculus; II, inner Integument; Mp, Micropyle; Nu, nucellus; OC, ovary cavity; OI, outer integument; OW, ovary wall; Pl, placenta; PT, pollen tube; Sp, septum.

## 4. Discussion

According to the classification by Heslop-Harrison and Shivanna [32] the Macabeo stigma is a wet type with multicellular uniseriate papillae. Although Bouard [33] claimed the presence of hermaphrodite and female flowers in some cultivated varieties (Macabeo, François Noir, and Malvoisie), we did not detect the presence of female flowers. The presence of female flowers cannot be ruled out as we have observed a limited number of flowers. The results showed that cv. Macabeo is self-fertile. Without any foreign pollen present, berries produced seeds, therefore, with pollen from self-plant. Traces of cleistogamy (*sensu lato*) have also been obtained for this cultivar. Pollen was observed on the emasculated flowers' stigmas. This pollen came from the anthers of their own flower before emasculation and, therefore, before capfall. The same result has been previously obtained for other cultivars where pollen also was observed on the stigma before capfall [21–23]. Ovule fertilization was observed in the emasculated flowers, which suggests that germination and pollen tube growth happened in a very early flower stage. These results corroborate with the fact presented by many *V. vinifera* cultivars studies that these are autogamous, and self-pollination often occurs before capfall [5,21].

Some unwanted pollen contamination could occur during emasculation. However, in this case, fewer seeds would be expected. The production of a similar seed number in both treatments with no significant differences made it difficult to attribute these seeds to an undesired process.

The cross-pollination treatment clearly increased the presence of pollen tubes growing in the stigmas and styles of the treated flowers. This result is consistent with previous studies that have supported the notion that cross-pollination improves fertilization and, in some varieties, improves fruit set [7,8,10,34]. Fertilization improvement was observed for cv. Macabeo, but was not necessary for fruit set as fruit set was enough and with no significant differences between the emasculated and

cross-pollinated flowers. It seems that ovule fertilization is successful even with little own pollen and therefore no foreign pollen is needed.

In the cross-pollination treatment, flowers were fixed only 24 h after pollination. By this time, profuse pollen tube growth was observed close to the micropyle. For some flowers, tubes were inside, but only in the first half of the stylar canals. For other flowers, tubes were at the end of stylar canals, very close to the ovule. These observations suggest that fertilization is very fast and can be as short as 24 h. No previous studies on the in vivo growth rate of pollen tubes have been found for *Vitis vinifera*; e.g., for mandarins (*Citrus clementina* Hort. ex Tanaka), tubes reached the end of stylar canals 8 to 10 days after pollination [35]. Pereira et al. (2018) reported the percentages of in vitro pollen germination of 15 *Vitis vinifera* L. cultivars, but not the growth rate. In our case, ovule fertilization for cv. Macabeo could take place at around 24 h after pollen arrival.

Finally, the in vivo ovule fertilization for cv. Macabeo is documented for the first time. Fluorescence microscopy images allowed us to observe the entrance of pollen tubes through the micropyle to the ovule. All this detailed information can be useful for breeding purposes.

## 5. Conclusions

The presented results on reproductive biology of economically important grape cultivar Macabeo can be useful for potential agronomic applications during flowering time or for breeding programs. This cultivar is autogamous and selfing was documented before the flower opens because pollen was found on the stigma surface before capfall and ovules were fertilized in emasculated flowers. Cross-pollination increased the amount of pollen tubes penetrating the style and ovary, but it had no effect on fruit set. Fertilization in this cultivar is fast; 24 h after pollination, the pollen tubes reach the end of stylar canal close to the entrance to the ovary and ovules.

**Supplementary Materials:** The following are available online at http://www.mdpi.com/2077-0472/10/12/647/s1, Figure S1: Pictures taken at the experimental site, Figure S2: Binocular view of a grapevine flower, Figure S3: Longitudinal toluidine blue-stained semi-thin section (1–2 μm) of a Macabeo flower.

**Author Contributions:** Conceptualization, H.M.; software, A.G.; validation, F.G.-B., J.R.A., A.G. and H.M.; formal analysis, A.G., F.G.-B., J.R.A. and H.M.; investigation, R.B., N.C., F.G.-B and J.R.A.; resources, F.G.-B., J.R.A. and H.M.; data curation, R.B. and A.G.; writing—original draft preparation, H.M. and F.G.-B.; writing—review and editing, F.G.-B., J.R.A., A.G., R.B. and H.M.; visualization, F.G.-B., J.R.A., A.G., R.B. and H.M.; supervision, H.M.; project administration, H.M.; funding acquisition, H.M. All authors have read and agreed to the published version of the manuscript.

**Funding:** This research was supported by the Asociación Club de Variedades Vegetales Protegidas as part of a project undertaken with the Universitat Politècnica de València (Spain, UPV 20190822), of which H. Merle was the principal researcher. There was no additional external funding received for this study.

**Acknowledgments:** The authors thank the farmer José Miguel Cebrián for providing technical assistance and for the orchard.

**Conflicts of Interest:** The authors declare no conflict of interest.

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
