# Peer review of "In Vivo Pollen Tube Growth and Evidence of Self-Pollination and Prefloral Anthesis in cv. Macabeo (Vitis vinifera L.)"

_agriculture, doi:10.3390/agriculture10120647_

Round 1

Reviewer 1 Report

I have now reviewed the manuscript entitled, “In vivo pollen tube growth and evidence of self-pollination and cleistogamy in cv. Macabeo (Vitis vinifera L.)”. The authors performed an emasculation and a cross-pollination treatment on flower clusters of two individual grape vines to demonstrate that cultivar Macabeo exhibits cleistogamous pollination and self-fertilization, in which pollen grains are dehisced from the anthers and begin germinating on the stigma before anthesis. Although the sample size is somewhat small, the work appears to be very carefully and thoroughly performed. This kind of careful natural history study is still much needed for many agricultural species.

I found the piece to be generally well communicated, although there are many instances of somewhat awkward phrasing, mostly involving extraneous or missing definite articles. I have flagged the few places in which awkward phrasing or other grammatical issues impeded understanding the point being communicated in my comments below.

One additional general comment I would like to make is that, while the authors have clearly communicated the results of their study, they have not made clear why it matters that they found what they did. What are the implications of their findings for the cultivation of Macabeo grapes, for the viticulture industry at large, and agriculture in general? Why should the readers care that Macabeo grapes are capable cleistogamy? It would be desirable for the authors to clearly communicate the implications and merit of their findings.

Specific comments are as follows; hopefully these will help improve the quality of the manuscript.

L36-38: Is it really relevant to discuss Vitis rotundifolia, a separate species (in a separate subgenus, no less), when the focus of the discussion at this point is on different cultivars of vitis vinifera?

L40-41: I am unable to understand the meaning of this sentence. Please revise.

L44: Please define “capfall”.

L50: Does this mean that Heazlewood and Wilson observed that no pollen tubes had grown? Or, that they did not undertake the process of observing pollen tubes? Please write unambiguously.

L51-52: The use of the word “where” here makes the sentence grammatically problematic, please revise.

L80: Does “S1” here refer to the appendix SM1? Please clarify.

L92-96: Were the flowers cut directly into FAA solution? Or were they cut, placed into empty vials, refrigerated, and then finally put into FAA solution when the designated amount of time has elapsed? If the latter, I would expect that refrigeration would potentially interfere with pollen tube growth. Please clarify the methodology here.

L125-128: I would recommend that the authors erect another section in Methods, “2.7. Statistical Analysis” to document the statistics that were performed, rather than mentioning them in the Results section.

L132-133: As with the above comment, please report exactly how the effect of replicate was analyzed, in a separate section reporting statistical analyses.

Figure 1: This is simply an aesthetic comment, but the text of the graph is very small relative to the graphic, making it difficult to read. Please increase the text size of legends and axis units. Although, after viewing Figure 2 and seeing the information presented in Table 1, it does not seem to me that Figure 1 is even needed.

Figure 2: Seeds are discrete quantities, so I’m not sure why the densities are plotted in continuous curves instead of discrete bars. The continuous curves give the illusion that the variable of interest is the area under the curve, whereas in truth, only the peaks of the curves at each discrete integer matters.

L147 and elsewhere: The authors interchangeably use “berry”, “fruit”, and “grape” in this manuscript. It may be desirable to use just one term to avoid confusion.

Figure 3: As in Figure 1, please enlarge the text.

L163: Formed by what? Do the authors simply mean “a film formed of secretory material”?

Figure 5, 6: “emasculate” is a verb, so please use “emasculated” to describe flowers.

Figure 7: Panel D is not identified in the legend.

L240: With only two individual grape vines examined for this study, it is not possible to rule out the absence of female flowers even for just the study site, let alone the cultivar.  

L256-258: It would be great if the authors could comment on the disconnection between the improvement in pollen tube growth but lack of improvement in fruit set and seed set.

L266-267: How can the authors rule out cleistogamous pollination in the case of the hand-pollinated treatment? Is there a possibility that all instances of pollen tubes having reached the ovule in the hand-pollinated treatment represent pollen tubes from cleistogamous self-pollen that germinated long before the application of outcross pollen? Please clarify.

L268-270: Please discuss the significance of observing in vivo ovule fertilization for this particular cultivar. Why is it important that this is documented?

SM3: It appears that TT was not labeled in the diagram despite appearing in the legend. Was this eschewed unintentionally?

Author Response

Reviewer #1:

We truthfully thank the reviewer #1 comments as these suggestions have clearly improved the manuscript. We have very carefully revised it after taking into account each comment. A point-by-point response to every requested change is provided below.

I have now reviewed the manuscript entitled, “In vivo pollen tube growth and evidence of self-pollination and cleistogamy in cv. Macabeo (Vitis vinifera L.)”. The authors performed an emasculation and a cross-pollination treatment on flower clusters of two individual grape vines to demonstrate that cultivar Macabeo exhibits cleistogamous pollination and self-fertilization, in which pollen grains are dehisced from the anthers and begin germinating on the stigma before anthesis. Although the sample size is somewhat small, the work appears to be very carefully and thoroughly performed. This kind of careful natural history study is still much needed for many agricultural species.

I found the piece to be generally well communicated, although there are many instances of somewhat awkward phrasing, mostly involving extraneous or missing definite articles. I have flagged the few places in which awkward phrasing or other grammatical issues impeded understanding the point being communicated in my comments below.

One additional general comment I would like to make is that, while the authors have clearly communicated the results of their study, they have not made clear why it matters that they found what they did. What are the implications of their findings for the cultivation of Macabeo grapes, for the viticulture industry at large, and agriculture in general? Why should the readers care that Macabeo grapes are capable cleistogamy? It would be desirable for the authors to clearly communicate the implications and merit of their findings.

Specific comments are as follows; hopefully these will help improve the quality of the manuscript.

Answer. We greatly appreciate the comments and suggestions of Reviewer # 1 and we believe that suggestions did serve to improve the manuscript. The detailed and meticulous review of the manuscript has been very useful.

In the conclusion section we included the following sentence to highlight the importance of the results:

Line 312         Having a greater knowledge of the reproductive biology of cv. Macabeo can be useful for potential agronomic applications during flowering time or for breeding programs.

L36-38: Is it really relevant to discuss Vitis rotundifolia, a separate species (in a separate subgenus, no less), when the focus of the discussion at this point is on different cultivars of vitis vinifera?

Answer. We appreciate the reviewer's suggestion. At this initial point of the “Introduction” we think it is interesting to know how the mating system in other subgenus is, to have a broader point of view. We would prefer to keep the reference.

L40-41: I am unable to understand the meaning of this sentence. Please revise.

Answer. The original sentence has been replaced by:

L41     How and when pollen is transferred from anthers to stigmas has been discussed for V. vinifera cultivars.

L44: Please define “capfall”.

Answer. The definition has been included in the paragraph as follows:

L45     The shedding of the calyptra is called capfall.

L50: Does this mean that Heazlewood and Wilson observed that no pollen tubes had grown? Or, that they did not undertake the process of observing pollen tubes? Please write unambiguously.

Answer. The paragraph has been rewritten to be unambiguous

L58     Heazlewood and Wilson (2004) confirmed the presence of pollen on the stigma of cv. Pinot Noir before capfall. By studying that pollen, these authors did not find any pollen tube growth before capfall. They concluded, conversely to Staudt, that fertilization should occur after the shedding of the calyptra.

L51-52: The use of the word “where” here makes the sentence grammatically problematic, please revise.

Answer. The word "where" has been replaced by the word "since" for a better understanding.

L63     Cleistogamy does not appear in all cultivars, since mechanisms may be present inside the cap to prevent it.

L80: Does “S1” here refer to the appendix SM1? Please clarify.

Answer. Yes. S1 has been replaced by SM1.

L92-96: Were the flowers cut directly into FAA solution? Or were they cut, placed into empty vials, refrigerated, and then finally put into FAA solution when the designated amount of time has elapsed? If the latter, I would expect that refrigeration would potentially interfere with pollen tube growth. Please clarify the methodology here.

Answer. Thanks for the comment. The methodology was not correctly explained. The flowers were cut and immediately introduced into a vial with the FAA solution. Then, flowers were transported in those vials to the laboratory, always maintaining the cold chain. Added the explanation:

L 105  Six flowers per grape cluster and treatment were collected for microscopy. All the flowers were sampled early in the morning and immediately introduced into a vial with FAA solution (10 ml formaldehyde 37%, 50 ml ethyl alcohol 95%, 5 ml glacial acetic acid; 35 ml of water). Sealed vials were kept in an insulated container to be transported to the laboratory, where they were refrigerated at 4ºC until analyzed. All the collected flowers were fixed with the FAA solution 48 h after calyptra removal.

L125-128: I would recommend that the authors erect another section in Methods, “2.7. Statistical Analysis” to document the statistics that were performed, rather than mentioning them in the Results section.

Answer. The previous "software" section has been replaced by "statistical analysis" where the details of the analyses performed have been included, as follows:

L 138  Statistical analyses and figures were made using R language [25] with RStudio [26]. Also, agricolae [27] and ggplot2 [28] packages were used for post-hoc analyses and graphs, respectively. The average, standard error, skew, and kurtosis were assessed per treatment. For the means comparisons, the ANOVA test did not meet the normality of residuals requirement and, therefore, the non-parametric Kruskal-Wallis test was used. Normality of residuals was tested using Shapiro–Wilk test. The effect of experiment repetition was checked by the non-parametric Kruskal-Wallis test, within the treatments between vines.

L132-133: As with the above comment, please report exactly how the effect of replicate was analyzed, in a separate section reporting statistical analyses.

Answer. This has been included in the section "statistical analysis"

Figure 1: This is simply an aesthetic comment, but the text of the graph is very small relative to the graphic, making it difficult to read. Please increase the text size of legends and axis units. Although, after viewing Figure 2 and seeing the information presented in Table 1, it does not seem to me that Figure 1 is even needed.

Answer. Thanks for the suggestion. We have increased the font size of the figures as suggested. Figure 1 is more focused on the means, while Figure 2 aims to focus on the distribution of data. We would like to keep both figures as we think both figures are interesting.

Figure 2: Seeds are discrete quantities, so I’m not sure why the densities are plotted in continuous curves instead of discrete bars. The continuous curves give the illusion that the variable of interest is the area under the curve, whereas in truth, only the peaks of the curves at each discrete integer matters.

Answer. We agree with Reviewer #1, he/she is right. The density graph has been replaced by a stacked barplot.

L147 and elsewhere: The authors interchangeably use “berry”, “fruit”, and “grape” in this manuscript. It may be desirable to use just one term to avoid confusion.

Answer. "berry" has been replaced by "grape" or by "fruit set" throughout the manuscript.

Figure 3: As in Figure 1, please enlarge the text.

Answer. Done, as for figure 1 and 2

L163: Formed by what? Do the authors simply mean “a film formed of secretory material”?

Answer. The reviewer's observation is correct, we mean “a film formed of secretory material”. It has been corrected in the manuscript (Line 188).

Figure 5, 6: “emasculate” is a verb, so please use “emasculated” to describe flowers.

Answer. Thanks for the comment, "emasculate" has been replaced by "emasculated"

Figure 7: Panel D is not identified in the legend.

Answer. The figure caption has been corrected as follows.

Line 252         Figure 7 Longitudinal blue violet autofluorescence-stained freeze section (20 μ m) of a cross-pollinated flower. a) Pistil with visible abundant germinating pollen grains (magnified stigma in b), pollen tubes penetrating transmission tissue of the style and inside the ovule; (c) Pistil with visible abundant germinating pollen grains (magnified stigma in d), reaching half of the stylar canal. Abbreviations. Ch, chalaza; Fu, funiculus; Ne, nectary; O, ovary; OC, ovary cavity; Ov, ovule; Pa, papillae; PG, pollen grain; Pl, placenta; PT, pollen tube; S, style; Sp, septum; St, stigma; TT, transmission tissue; VB, vascular bundle.

L240: With only two individual grape vines examined for this study, it is not possible to rule out the absence of female flowers even for just the study site, let alone the cultivar.  

Answer. We did not mean that there are no female flowers in Macabeo cultivar, only that in our experiment and observations we did not found them.

Therefore, we have rewritten that part as follows:

Line 274 … we did not detect the presence of female flowers. The presence of female flowers cannot be ruled out as we have observed a limited number of flowers.

L256-258: It would be great if the authors could comment on the disconnection between the improvement in pollen tube growth but lack of improvement in fruit set and seed set.

Answer. We have added the following sentence to make it clearer:

Line 295         It seems that ovule fertilization is successful even with little own pollen and therefore no foreign pollen is needed.

L266-267: How can the authors rule out cleistogamous pollination in the case of the hand-pollinated treatment? Is there a possibility that all instances of pollen tubes having reached the ovule in the hand-pollinated treatment represent pollen tubes from cleistogamous self-pollen that germinated long before the application of outcross pollen? Please clarify.

Answer. Cleistogamous pollination in the case of the hand-pollinated treatment cannot be completely rule out. However, due to the stamens position and the anthers maturity, most of the pollen came from brushing with the foreign anthers.

In any case, a small percentage of cleistogamy even in this hand-pollinated treatment cannot be ruled out. This did not substantially influence the results since the advance of the profuse mass of pollen tubes can be clearly observed in the hand-pollinated flowers, the tubes being close to the ovules (as a consequence of pollen release by brushing) ; but this was not found in only emasculated flowers.

L268-270: Please discuss the significance of observing in vivo ovule fertilization for this particular cultivar. Why is it important that this is documented?

Answer. Having a greater knowledge of the reproductive biology of any cultivated species and its cultivars can be useful for potential agronomic applications during flowering time e.g., or for breeding programs. Therefore, the following sentence has been added

L308   All this detailed information can be useful for breeding purposes.

SM3: It appears that TT was not labeled in the diagram despite appearing in the legend. Was this eschewed unintentionally?

Answer. It was a mistake. We have removed TT from the figure caption

Thank you for your review.

Reviewer 2 Report

Dear authors,

I have only detected some minor corrections to the text which are included in the pdf document (see attached). Keep the good work. All the best.

Author Response

Reviewer #2

We truthfully thank the reviewer #2 comments as these suggestions have clearly improved the manuscript. We have very carefully revised it after taking into account each comment. A point-by-point response to every requested change is provided below.

Dear authors,

I have only detected some minor corrections to the text which are included in the pdf document (see attached). Keep the good work. All the best.

Answer. Dear reviewer, we greatly appreciate the minor corrections. We have followed all them to improve the manuscript.

L48      Please rephrase this sentence.

Answer. The sentence has been rewritten as follows:

L58     Heazlewood and Wilson (2004) confirmed the presence of pollen on the stigma of cv. Pinot Noir before capfall. By studying that pollen, these authors did not find any pollen tube growth before capfall. They concluded, conversely to Staudt, that fertilization should occur after the shedding of the calyptra.

L53      Please include reference (s) that corroborates this affirmation. Please, again, include references or indicate which are the existing few studies.

Answer. References have been included

L 247   Please rephrase. These results corroborate with the fact presented by many V. vinifera cultivars studies that these are autogamous, and self-pollination often occurs before capfall.

Answer. The sentence has been rewritten as suggested by the reviewer 2 (Line 284).

All spelling and punctuation issues suggested by the reviewer have been fixed.

Thank you very much for your review.

Reviewer 3 Report

In vivo pollen tube growth and evidence of self-pollination and cleistogamy in cv. Macabeo (Vitis vinifera L.)

The paper is interesting, documentation of self- and cross pollination in Vitis vinifera L. cv. Macabeo is very good quality. The main objection is the term cleistogamy used in relation to cv Macabeo flowers. Cleistogamy is precisely defined (Uphof 1938; Lord 1981; Culley and Klooster 2007) and the incidentally probably bud-pollinated flowers of grape cannot be described as cleistogamous. Bud-pollination is well known phenomenon in e.g., Brassicaceae explaining as self-pollination is possible in self-incompatible species because at early bud development the stigma surface not yet accumulate factors sufficient to inhibit self-pollen germination and pollen tube growth.

Detailed Reviewer’s comments for Authors are in attached file

Author Response

Reviewer #3:

We truthfully thank the reviewer #3 comments as these suggestions have clearly improved the manuscript. We have very carefully revised it after taking into account each comment. A point-by-point response to every requested change is provided below.

In vivo pollen tube growth and evidence of self-pollination and cleistogamy in cv. Macabeo (Vitis vinifera L.)

The paper is interesting, documentation of self- and cross pollination in Vitis vinifera L. cv. Macabeo is very good quality. The main objection is the term cleistogamy used in relation to cv Macabeo flowers. Cleistogamy is precisely defined (Uphof 1938; Lord 1981; Culley and Klooster 2007) and the incidentally probably bud-pollinated flowers of grape cannot be described as cleistogamous. Bud-pollination is well known phenomenon in e.g., Brassicaceae explaining as self-pollination is possible in self-incompatible species because at early bud development the stigma surface not yet accumulate factors sufficient to inhibit self-pollen germination and pollen tube growth.

According to cleistogamy classification by Culley and Klooster (2007), three main types of cleistogamy were recognized based on developmental pathways: dimorphic, complete, induced. Dimorphic cleistogamy (= true cleistogamy according to Lord’s, 1981 classification) is defined as formation of chasmogamous, open, cross-pollinated flowers (CH) and cleistogamous, obligate self-pollinated flowers (CL) on one individual, differing in floral morphology. Complete cleistogamy, formation of exclusively CL flowers by a plant, was observed and documented in grasses and orchids. Induced cleistogamy (=pseudocleistogamy according to Lord, 1981; = ecological cleistogamy based on old Uphof's, 1938 classification) is influenced by environmental conditions (drought, low or high temperature, light, nutrition, surrounding soil). Both flower types (CH and CL) differ in their morphology, CL flowers has reduced perianth and modified stamens and pistil adapted to self-pollination (e.g., Viola uliginosa, MaĹ‚obÄ™cki et al. 2016; Botanical Journal of the Linnean Society, 2016,182, 180–194).

Authors named grape flowers cleistogamous not citing even review papers on cleistogamy. The text is written in poor English and needs to be corrected for the style and grammar by the native speaker. Figure captions are complicated and need correction.

The novelty of this paper: there are several papers on mating system of different cultivars of Vitis vinifera indicating that they are self-compatible. The novelty is the new cultivar of grape and very good documentation confirming self- and cross compatibility of this cultivar.

Detailed Reviewer’s comments for Authors:

ABSTRACT:

Lines 22-24: The results showed that cv. Macabeo is self-fertile and most likely cleistogamous.

Reviewer’s comments: Cleistogamy is a widespread phenomenon in the angiosperms, occurring in 50 families of monocots and dicots (Uphof 1938; Lord 1981; Culley and Klooster 2007). Cleistogamous species are able to develop open, cross-pollinating chasmogamous (CH) flowers with typically formed organs, and reduced, closed, obligatory self-pollinating cleistogamous (CL) flowers. Cleistogamy based on definition refers to the presence of two flower morphs on a single plant. Authors should use the term cleistogagamous species/variety with caution when two types of flowers are not formed by a single plant temporarily or simultaneously.

Answer. We have used the term "cleistogamy" sensu lato. This term has been frequently used in previous studies for Vitis vinifera  in sensu lato, as “selfing before flower opens”.

Some examples with the original sentences are (all referring to Vitis):

Staudt (1999)

“The two cultivars were partially cleistogamous”

“Flowers of both cultivars of Vitis vinifera tended to pre-floral anthesis, they were partially cleistogamous.”

Sampson et al., 2001 and Mullins et al., 1992

“Selfing ensures some level of fruit set in grapes and can occur before buds open (cleistogamy)”

“Perfect flowers of muscadine grapes readily selfed; cleistogamy is a possible mechanism for pollen transfer because 22% of pistils exhibited peroxidase activity”

“We have some preliminary evidence for cleistogamy in muscadines, a form of selfpollination that occurs in other grape species (Mullins et al., 1992).”

“Even cleistogamy does not preclude crosspollination for muscadine flowers.”

Vasconcelos et al., 2009

“The most recent views are that self-pollination is important and often happens before capfall (cleistogamy) and that cross-pollination also occurs and often results in better seed set in the berries”

Muller-Thurgau were pollinated before opening and the growth of pollen tubes had already started, concll1- ding that the two cultivars were partially cleistogamous

Therefore, we have included in the Introduction section review papers on cleistogamy as suggested, the sensu lato of the term used in this study, and we have reduced the use of the word "Cleistogamy" replacing it with “prefloral pollination” and/or “selfing occurs before flower opens” when appropriate.

Line 34    True cleistogamy is defined as the formation of chasmogamous, open, cross-pollinated flowers and cleistogamous, obligate self-pollinated flowers on one individual, differing in floral morphology  [19,20]. In Vitis, the term has been used sensu lato to refer to flowers that are pollinated and fertilized before flower opens, but without specific perianth, stamens and pistil modifications [5,8,21]. In this study the terms "cleistogamous” and “cleistogamy” are used to refer self-pollinated flowers prior to capfall, but not necessarily “true cleistogamy”.

Lines: 25-26: Pollen was found over the stigma of flowers before capfall and ovule fertilization was observed even in emasculated flowers; which suggests that germination and pollen tube growth happened in a very early flower development stage.…..ovule fertilization was observed even in emasculated flowers

 this part of the sentence is incomprehensible. If flower was emasculated how pollen could be on stigma and fertilized the ovule (rather an egg cell)?

Answer. Emasculation was carried out 24 hours before the natural capfall. We observed pollen on the stigma of those emasculated flowers, which is indicating that this pollen probably arrived at the stigma before the emasculation was performed.

The release of pollen to the stigma before capfall has been demonstrated for several Vitis vinifera cultivars.

  1. INTRODUCTION

Lines: 45-52: This part cited papers on mating system in different grape cultivars repeating that some cultivars are cleistogamous. The description rather refers to bud-pollination. This should be discussed – cleistogamy or bud-pollination?

Answer. The original papers used the term “cleistogamy” not “bud-pollination”. These papers have probably used the term "cleistogamy" in its sensu lato, that is, selfing before flower opens.

The original sentences are:

Staudt (1999)

The two cultivars were partially cleistogamous

Flowers of both cultivars of Vitis vinifera tended to pre-floral anthesis, they were partially cleistogamous.

Lines: 55-57: No information was found on the time that pollen takes from germination to ovule fertilization and, to our knowledge, no scientific analysis of the mating system of cv. Macabeo has been published.

Reviewer’s comments: This sentence should be re-written, e.g., “To our knowledge there is no information about the time needed for pollen tube from pollen germination on the stigma to enter into the ovule to fertilize the egg cell”

Answer. Thank you very much for your suggestion. The sentence has been rewritten as suggested by the reviewer as follows:

Line 66           To our knowledge there is no information about the mating system of cv. Macabeo or the time needed for pollen tube from pollen germination on the stigma to enter into the ovule to fertilize the egg cell.

In the Introduction section the terms cleistogamy and bud-pollination should be briefly described along with citations.

Answer. Done in line 34.

  1. MATERIALS AND METHODS

1.1.Experimental Site

Line 66: SM1a should be cited

Answer. SM1a has been cited as suggested.

L76     Casas Ibañez (Albacete, Spain, 39.331037 N, -1.478663 W) (SM1a).

2.2.Treatments

Line 77: SM1b,c should be cited

Answer. SM1b,c has been cited as suggested.

L90     … without calyptra were removed immediately before treatment (SM1b,c).

Line 82: SMd should be cited

Answer. SM1d has been cited as suggested.

L95     … treatment and were never opened before evaluations (SM1d).

2.4.Flower Sampling for Microscopy and 2.5. Microscopy should be combine in one section.

Answer. Both sections have been combined as suggested.

  1. RESULTS

3.1.Analysis of the Mean Seed Number per Grape and berry Set

The title should be modify: Mean seed number per grape and berry set

Answer. We have followed the journal style, according to the template.

Lines 125-128: Should be removed as it is repetition of Material and Methods (Statistic)

Answer. Lines 125-128 have been removed as suggested by the reviewer.

Line 130: (Figure 1; Table 1; Kruskal-Wallis p value = 0.811);cite only Figure and Table (Figure 1; Table 1) as other information are in Figure 1 caption and in foot note of and Table.

Answer. We appreciate the reviewer's suggestion. We have removed the reference to the KW test.

Line 154         … was found between treatments (Figure 1; Table 1).

Lines 123-133: The experiment was repeated? There is no information about repetition in Materials and Methods.

Authors should reorganized Figures e.g., Figure 1 and Figure 2 should be combine in one plate as they concern the same problem. Fig. 1a and 1b.

Answer. We apologize for this misunderstanding. Each vine was an entire repetition. We have modified the Materials and Methods section for better understanding:

Line 81           For the experiments, six grape bunches were randomly selected. The entire experiment was repeated in two healthy vines. Both vines were treated separately with just a few days difference.

Figure 2 has been replaced by a clearer figure. Figure 1 focuses on means, while Figure 2 focuses on the distribution of the data. We would like to keep both separately.

Lines 147-157: This part should be removed from the manuscript because the experiment was not well planned and executed.

Answer. Although the experimental design was not intended to evaluate fruit-set, some preliminary results can be drawn by properly noting their limitation.

Berry-set was similar in both treatments without significant differences with a similar number of treated flowers. Although we cannot state this outright, we can suggest that it appears that emasculation did not affect fruit-set. That is why we include the following sentence: Therefore, the result should be cautiously taken.

Line 160: Pollen diameter was 20 μ m (S3). There is lack of information in Material & Methods section how many pollen grains were measured to estimate the diameter of pollen grains.

Answer. We have added that detail in the materials and methods section as follows:

Line 132         Ten pollen grains were measured to estimate the diameter of pollen grains through toluidine blue-stained semi-thin section (SM3b).

Lines 161-169: In this part Figure 4 and S3 are cited. Fig. 4 and S3 should be combine as one plate because some photos are repetition (e.g. S3a). It would be easier to analyze flower structure on one plate with selected important detailed elements.

Answer. Both figures are similar but from different flowers and in each of them different magnified areas are displayed. To make a combined figure, the size would have to be reduced, which would be detrimental to structures observation. For a good visualization we cannot include so much information in a single figure. Essential information is in Figure 4. Figure SM3 only includes additional information. We would like to keep them separate since the size is optimal for their correct exposition.

Lines: 188-189: Did Authors observe that the anthers were open when flower was emasculated in the stage before capfall?

Answer. The anthers were fully developed, not completely open but probably some at least partially open.

Line 210: Transmission tissue add abbreviation (TT)

Answer. The abbreviation has been added as suggested.

Lines: 211-212: The sentence: “We have observed flowers with the mass of pollen tubes at the base of stylar canals (Figure7a) and flowers with the mass of pollen tubes at the first half of the TT” should be re-formulate: e.g.: In some flowers the mass of pollen tubes reached the base of stylar canal (Fig. 7a), in other flowers the pathway of pollen tubes was shorter finishing in the middle of the stylar canal (Fig. 7c).

Answer. Thank you very much for your suggestion. The sentence has been rewritten as suggested by the reviewer as follows:

L236   In some flowers the mass of pollen tubes reached the base of stylar canal (Figure 7a), while in other flowers the pathway of pollen tubes was shorter finishing in the middle of the stylar canal (Figure 7c).

Line: 213: “…pollen tubes entered through the micropyle and caused ovule fertilization”

Should be more precise description: “pollen tubes entered through the micropyle into the ovule, releases sperm cells to fertilize an egg cell and the central cell of female gametophyte”

Answer. we appreciate the suggestion. The sentence has been rewritten as indicated by the reviewer as follows:

L239   In some pollinated flowers, pollen tubes entered through the micropyle into the ovule, releases sperm cells to fertilize an egg cell and the central cell of female gametophyte (Figure 8).

Lines: 217-220: This part should be moved to the Discussion section and the text should be reformulated.

Answer. The text has been reformulated to improve it. At this point, we prefer to highlight these results here. Later in the discussion sections, these results are analyzed in context with other cited studies.

Line 245         This observations are pointing to a very fast fertilization process as the mass of tubes reached the vicinity of the ovary in only 24 h, and to a fertilization improvement due to a significant increase in the number of pollen tubes growing inside the flower.

  1. DISCUSSION

Lines 241-249: See above comments on cleistogamy

Answer. This point has been clarified in the introduction.

Lines: 259-263 and 268-260: These parts are repetition of the results even figures are cited, should be re-written.

Answer. References to figures have been removed. Part of the discussion has been modified. In the discussion section, the results of this study are discussed and compared with the results obtained in previous studies on Vitis vinifera.

  1. CONCLUSION

This section is not Conclusion but Summary and should be re-written

Answer. At least in somehow, a conclusion consists of briefly summarizing the most relevant points and highlighting why these results are important. Conclusion should not include new information that has not been mentioned before. Simply emphasize the most important results as clearly as possible. We have modified the conclusions to be clearer and to avoid redundancies.

Line 307         Having a greater knowledge of the reproductive biology of cv. Macabeo can be useful for potential agronomic applications during flowering time or for breeding programs. In this study we have shown that Macabeo cultivar is autogamous and probably, selfing occurs before flower opens. Pollen was found on the stigma surface before capfall and ovule fertilization was observed even in the emasculated flowers. Cross-pollination increased the amount of pollen tubes growing inside flowers, but this increase was unnecessary for fruit set. Pollen tube growth was very fast, and 24 h were enough to arrive to the vicinity of the ovule .

FIGURES:

1) There is inconsistency in the labeling of the figures and in the description of the figures. The photos are labelled with small letters, in the figures captions with large letters or small letters.

Answer. The reviewer is right. We have changed all figure captions to lowercase letters.

2) In Figure 7 caption there is lack of photos (d) description, this photos is not cited in the text.

Answer. We apologize for the mistake. The complete figure caption has been included.

3) Description of figures should be more compact and more clear and not too descriptive e.g. the description of Figure SM3: “Longitudinal toluidine blue-stained semi-thin section (1-2 μm) of a Macabeo flower. a) Flower structure, the details (red square marking) are magnified: papillae on stigma with visible secretion covering cell wall and pollen grains (b), raphids in cells of style (c), style epidermis, subepidermal tissue with druses, and the parenchymal cells (d). Abbreviations. Cu, cuticle; D, discus; Dr, druses; Ep, epidermis; F, filament; Ne, nectary; O, ovary; OC, ovary cavity; Ov, ovule; OW, ovary wall; P, peduncle; Se, septum; Pa, papillae; PG, pollen grain; PL; pellicle layer; Ra, raphide; S, style; SC, stylar cells;. Se, sepals; Sp, septum; St,stigma; TT, transmission tissue; VB, vascular bundle.

Answer. We appreciate the reviewer's suggestion. We have modified the figure caption as suggested by the reviewer

4) Fig.SM2: new caption: Macabeo flower at different stages of development. (a) Immature flower with petals forming a cap or calyptra. (b) Flower with folded petals forming a calyptra at the top. (c) Flower without the calyptra removed by stamens, anthers at anthesis. Abbreviations: An, anther; Ca, calyptra; D, discus; F, filament; Ne, nectary; O, ovary; St, stigma.

Answer. Thanks for the comment. We appreciate the effort to improve the figure caption. We have modified the figure caption as suggested by the reviewer

5) Figure 7 description is not adequate to the photos on the figure: Longitudinal blue violet autofluorescence-stained freeze section (20 μ m) of a cross-pollinated flower. (a) At pistil stigma visible abundant germinating pollen grains (magnified in b), pollen tubes penetrating transmission tissue of the style and inside the ovule. (c) Detail of the fertilization process of a mature ovule. The entrance of a pollen tube in the embryo sac through the micropyle. Lack of photos (d) description. Abbreviations. Ch, chalaza; Fu, funiculus; Ne, nectary; O, ovary; OC, ovary cavity; Ov, 227ovule; Pa, papillae; PG, pollen grain; Pl, placenta; PT, pollen tube; S, style; Sp, septum; St, stigma; TT, 228transmission tissue; VB, vascular bundle.

Answer. Thanks for the comment. We have improved the figure caption as suggested.

Line 246         Figure 7 Longitudinal blue violet autofluorescence-stained freeze section (20 μ m) of a cross-pollinated flower. a) Pistil with visible abundant germinating pollen grains (magnified stigma in b), pollen tubes penetrating transmission tissue of the style and inside the ovule; (c) Pistil with visible abundant germinating pollen grains (magnified stigma in d), reaching half of the stylar canal. Abbreviations. Ch, chalaza; Fu, funiculus; Ne, nectary; O, ovary; OC, ovary cavity; Ov, ovule; Pa, papillae; PG, pollen grain; Pl, placenta; PT, pollen tube; S, style; Sp, septum; St, stigma; TT, transmission tissue; VB, vascular bundle.

Round 2

Reviewer 3 Report

The revised manuscript was improved but still needs correction. The section Material and Method should be shortened to avoid repetition. Figure captions should be corrected. Pollen diameter based on measurement of 10 pollen grains on sectioned material should be removed because insufficient sample size. Conclusion should be re-organized.In reviewer’s opinion the text needs English correction for style and grammar. 

Detail comments for Authors are included into attached document.

Author Response

Reviewer #3 (Round 2):

We welcome the reviewer's suggestions. Comments have been very helpful improving the manuscript.

The revised manuscript was improved but still needs correction. The section Material and

Method should be shortened to avoid repetition. Figure captions should be corrected. Pollen

diameter based on measurement of 10 pollen grains on sectioned material should be

removed because insufficient sample size. Conclusion should be re-organized.

In reviewer’s opinion the text needs English correction for style and grammar.

Reviewer’s comments for Authors

Line 104: 2.4. Flower Sampling and Microscopy

Rev. comment: The title of the section should be modify:

Flower Sampling for Pollination Experiments and Floral Structure Analysis

Answer. The title of the section has been modified as suggested by the reviewer.

L99      2.4. Flower Sampling for Pollination Experiments and Floral Structure Analysis

Lines 105-137: The all part of Material and Methods should be more concise without repetition. Suggestion is to re-organize this section as e.g. below

Six flowers per grape cluster and treatment were collected 48 h after calyptra removal. For cross-pollination experiment flowers were fixed 24 h after pollination, for emasculation treatment 48 h after anthers removal. All flowers were sampled early in the morning and immediately fixed in FAA solution (10 ml 106 formaldehyde 37%, 50 ml ethyl alcohol 95%, 5 ml glacial acetic acid; 35 ml of water), vials with fixed material were transported to the laboratory in an insulated container, next stored at 4ºC in the refrigerator until analyzed. For semi-thin sectioning, flowers were fixed in 2% Karnovsky fixative for 2 h at 4ºC before they were washed 3 times with 0.01 M PBS, pH 7.4, for 15 min each. Samples were dehydrated at room temperature in graded series of ethanol, starting at 50% and increasing to 70%, 95% and 100% for no less than 20–30 min at each step, embedded in Spurr’s resin according to the manufacturer’s instructions. Flowers were sectioned on 1–2 μm using a diamond knife (DIATOME Histo 45) and an ultramicrotome (Ultratome Nova LKB Bromma). Sections were stained with 0.1% toluidine blue, and examined with light microscopy (LM) and epifluorescence microscopy (EFM) to analyze stigma, style, ovary, ovules, pollen grains on stigma.

For flower morphology observation, a Leica M165 stereomicroscope with a high-resolution IC80HD digital image capture system controlled by the LAS program was used.

For flower structure analysis, flowers were sectioned longitudinally on ca. 15 μm with a freezing microtome (Leica CM 1325), sections were stained with aniline blue fluorochrome (0.1% in PBS buffer) for 15 min, subsequently washed with distilled water and mounted for observation under an epifluorescence microscope.

Ten pollen grains were measured to estimate the diameter of pollen grains through toluidine blue-stained semi-thin section (Figure S3b). All the LM and EFM observations were made by an Olympus Provis AX 70 fluorescence microscope equipped with an Infinity 2–3 C Lumenera® digital camera and analyzed with the “Infinity Analyze” software, v.6.4.1. For fluorescence microscopy, an Olympus U-ULS 100 HG epifluorescence system with a U-MWBV cube (excitation filter 400–440 nm, dichroic mirror 455 nm, barrier filter 475 nm) was used.

Answer. Thank you very much for your suggestion.

-. References to pollen diameter have been removed from Material and Methods and from results section, as suggested by the reviewer.

-. The entire materials and methods section has been improved based on the reviser's corrections (see the entire section).

Six flowers per grape cluster and treatment were collected 48 h after calyptra removal. For cross-pollination experiment flowers were fixed 24 h after pollination, for emasculation treatment 48 h after anthers removal. All flowers were sampled early in the morning and immediately fixed in FAA solution (10 ml 106 formaldehyde 37%, 50 ml ethyl alcohol 95%, 5 ml glacial acetic acid; 35 ml of water) or Karnovsky [27]. Vials with fixed material were transported to the laboratory in an insulated container, next stored at 4ºC in the refrigerator until analyzed.

For autofluorescence-stained sections, flowers were fixed in FAA solution and were sectioned longitudinally on ca. 20 μm with a freezing microtome (Leica CM 1325), sections were stained with aniline blue fluorochrome (0.1% in PBS buffer) for 15 min, subsequently washed with distilled water and mounted in a microscope slide for observation under an epifluorescence microscope.

For semi-thin sectioning, flowers were fixed in 2% Karnovsky fixative for 2 h at 4ºC before they were washed 3 times with 0.01 M PBS, pH 7.4, for 15 min each [27]. Samples were dehydrated at room temperature in graded series of ethanol, starting at 50% and increasing to 70%, 95% and 100% for no less than 20–30 min at each step, embedded in Spurr’s resin according to the manufacturer’s instructions. Flowers were sectioned on 1–2 μm using a diamond knife (DIATOME Histo 45) and an ultramicrotome (Ultratome Nova LKB Bromma). Sections were stained with 0.1% toluidine blue.

Sections were examined with light microscopy (LM) and epifluorescence microscopy (EFM) to analyze stigma, style, ovary, ovules, pollen grains on stigma. All the LM and EFM observations were made by an Olympus Provis AX 70 fluorescence microscope equipped with an Infinity 2–3 C Lumenera® digital camera and analyzed with the “Infinity Analyze” software, v.6.4.1. For fluorescence microscopy, an Olympus U-ULS 100 HG epifluorescence system with a U-MWBV cube (excitation filter 400–440 nm, dichroic mirror 455 nm, barrier filter 475 nm) was used. For flower morphology observation, a Leica M165 stereomicroscope with a high-resolution IC80HD digital image capture system controlled by the LAS program was used.

Lines 202-205: Figure caption. The style should be similar as in S3. It is not necessary to write the detail because it is evident on the photos. Suggestion is to describe figures in more concise manner.

Figure 4. Longitudinal toluidine blue-stained semi-thin section (1-2 μm) of a Macabeo flower. a) Flower structure, the details (red squares marking) are magnified: b) papillae on stigma and transmission tissue. c) Pollen tube growing through the transmission tissue. d) Ovule micropyle zone. ………

Answer. The figure caption has been corrected as suggested by the reviewer

Figure 4. Longitudinal toluidine blue-stained semi-thin section (1-2 μm) of a Macabeo flower. a) Flower structure, the details (red squares marking) are magnified: b) papillae on stigma and transmission tissue. c) Pollen tube growing through the transmission tissue. d) Ovule micropyle zone. Abbreviations. Ch, chalaza; Ep, epidermis; F, filament; Fu, funicule; II, inner integument; IS, intercellular space; Mp, micropyle; Ne, nectary; Nu, nucellus; O, ovary; OC, ovary cavity; OI, outer integument; Ov, ovule; P, peduncle; Pa, papillae; Pl, placenta; PT, pollen tube; Ra, raphide; S, style; Se, sepal; Sp, septum; St, stigma; TT, transmission tissue; VB, vascular bundle.

Line 219 Figure 5 captions

In figure description is 20 μm, in Material and Methods ca. 15 μm, should be corrected

Answer. It has been fixed in materials and methods as follows

…and were sectioned longitudinally on ca. 20 μm with a freezing microtome (Leica CM 1325),…

Lines 220-222 Figure 5 captions

Figure 5. Longitudinal blue violet autofluorescence-stained freeze section (20 μm) of emasculated flower. a) Flower structure, visible pistil with stigma and style, ovary with ovules. b) Papillous stigma, visible ungerminated pollen grains. c) Germinating pollen grains, pollen tubes on stigma and in style. d) Pollen tubes reach the stylar canals base. …….

Answer. The figure caption has been corrected as suggested by the reviewer

Figure 5. Longitudinal blue violet autofluorescence-stained freeze section (20 µm) of an emasculated flower. a) Flower structure, visible pistil with stigma and style, ovary with ovules. b) Papillous stigma, visible ungerminated pollen grains. c) Germinating pollen grains, pollen tubes on stigma and in style. d) Pollen tubes reach the stylar canals base. Abbreviations. D, discus; F, filament; Ne, nectary; O, ovary; OC, ovary cavity; Ov, ovule; OW, ovary wall; P, peduncle; Pa, papillae; PG, pollen grain; PT, pollen tube; S, style; Se, sepal; Sp, septum; St, stigma; TT, transmission tissue; VB, vascular bundle.

Lines 228-229: Figure 6 captions

Figure 6. Longitudinal blue violet autofluorescence-stained freeze section (20 μm) of an emasculated flower. a) Two mature ovules inside the ovary. b) Pollen tube penetrating through the micropyle into the embryo sac.

Please add in the text: embryo sac (=female gametophyte) and uniformly name in figure captions embryo sac or female gametophyte.

Answer. The figure caption has been corrected as suggested by the reviewer

Figure 6. Longitudinal blue violet autofluorescence-stained freeze section (20 µm) of an emasculated flower. a) Two mature ovules inside the ovary. b) Pollen tube penetrating through the micropyle into the embryo sac. Abbreviations. Ch, chalaza; Fu, funicule; II, inner Integument; Mp, Micropyle; Nu, nucellus; O, ovary; OC, ovary cavity; OI, outer integument; Ov, ovule; Pl, placenta; PT, pollen tube; Sp, septum; VB, vascular bundle.

Female gametophyte (=embryo sac) has been included in the text (Line 256).

Lines 252-255. Some correction of figure 7 captions Figure 7. Longitudinal blue violet autofluorescence-stained freeze section (20 μm) of a cross-pollinated flower. a) Pistil with visible abundant germinating pollen grains (magnified stigma in b), pollen tubes penetrating transmission tissue of the style and inside the ovule. (c) Pistil with visible abundant germinating pollen grains (magnified stigma in d), pollen tubes reaching half of the stylar canal.

Answer. The figure caption has been corrected as suggested by the reviewer

Figure 7. Longitudinal blue violet autofluorescence-stained freeze section (20 μm) of a cross-pollinated flower. a) Pistil with visible abundant germinating pollen grains (magnified stigma in b), pollen tubes penetrating transmission tissue of the style and inside the ovule. (c) Pistil with visible abundant germinating pollen grains (magnified stigma in d), most pollen tubes reaching half of the stylar canal. Abbreviations. Ch, chalaza; Fu, funiculus; Ne, nectary; O, ovary; OC, ovary cavity; Ov, ovule; Pa, papillae; PG, pollen grain; Pl, placenta; PT, pollen tube; S, style; Sp, septum; St, stigma; TT, transmission tissue; VB, vascular bundle.

“most” has been added to …c) most pollen tubes reaching half of the stylar canal. in contrast to a)

Lines 311-318: Conclusion

Please, re-write the conclusion e.g.:

The presented results on reproductive biology of economically important grape cultivar Macabeo can be useful for potential agronomic applications during flowering time or for breeding programs.

1) This cultivar is autogamous and selfing was documented before the flower opens because pollen was found on the stigma surface before capfall and ovules were fertilized in emasculated flowers.

2) Self-fertilization is beneficial when it is necessary to replicate a line of grapes with the desired characteristics.

3) Cross-pollination increased the amount of pollen tubes penetrating style and ovary, but it had no effect on fruit set.

4) Fertilization in this cultivar is fast, 24 h after pollination the pollen tubes reach the end of stylar canal close to the enter to the ovary and ovules.

Answer. Conclusion has been corrected as suggested by the reviewer. We prefer not to use numbers.

The presented results on reproductive biology of economically important grape cultivar Macabeo can be useful for potential agronomic applications during flowering time or for breeding programs. This cultivar is autogamous and selfing was documented before the flower opens because pollen was found on the stigma surface before capfall and ovules were fertilized in emasculated flowers. Cross-pollination increased the amount of pollen tubes penetrating style and ovary, but it had no effect on fruit set. Fertilization in this cultivar is fast, 24 h after pollination the pollen tubes reach the end of stylar canal close to the enter to the ovary and ovules.

We appreciate the effort made by the reviewer 3 that has allowed us to improve the manuscript.
